# Treatment Modalities for Non-Muscle Invasive Bladder Cancer: An Updated Review

**DOI:** 10.3390/cancers16101843

**Published:** 2024-05-11

**Authors:** Shannon McNall, Kailey Hooper, Travis Sullivan, Kimberly Rieger-Christ, Matthew Clements

**Affiliations:** 1Department of Urology, Lahey Hospital & Medical Center, Burlington, MA 01805, USA; kimberly.r.christ@lahey.org (K.R.-C.); matthew.clements@lahey.org (M.C.); 2Department of Translational Research, Lahey Hospital & Medical Center, Burlington, MA 01805, USA; kailey.e.hooper@lahey.org (K.H.); travis.b.sullivan@lahey.org (T.S.)

**Keywords:** non-muscle invasive, bladder cancer, clinical trial, Bacillus Calmette–Guérin

## Abstract

**Simple Summary:**

The mainstay of treatment for non-muscle invasive bladder cancer is transurethral resection of the tumor followed by intravesical treatment for intermediate- or high-risk disease. Duration and type of intravesical treatment depend on specific characteristics of the tumor and the patient’s disease course. Patients with non-muscle invasive bladder cancer are closely monitored with frequent surveillance cystoscopies following treatment. This type of cancer has a low progression rate; however, it has a high rate of recurrence with the current recommended treatments. Therefore, patients often find themselves having to undergo multiple resections and rounds of intravesical treatment throughout their lifetime. This is not only burdensome for patients but also costly for the healthcare system. Given the high recurrence with the standard treatment options, it is important to continue investigating novel therapies. The goal of this review article is to outline the most recent therapies available or in clinical trials for the treatment of non-muscle invasive bladder cancer.

**Abstract:**

The landscape of treatment for non-muscle invasive bladder cancer is rapidly changing. A complete and careful transurethral resection is the mainstay of initial treatment and is followed by intravesical therapy in intermediate or high-risk cases. The standard of care is intravesical BCG. Many alternative or additive approaches to this are being explored. We divided this review into three relevant spaces to consider these novel treatment approaches: (1) low-risk disease, for which intravesical therapy is not usually considered, (2) BCG-naïve disease (i.e., considering alternatives to the standard therapy), and (3) BCG-unresponsive disease. We performed a review of published literature and summarized ongoing trials in the United States. Novel approaches that we explored include surgical techniques for resection, alterations in dwell time for intravesical therapy, delivery method and schedule of intravesical therapies, new intravesical therapy agents, and systemic therapies (especially immunotherapy). These are thoroughly outlined throughout this review article, and the numerous modalities being studied demonstrate significant promise for the future treatment of the expanding space of NMIBC.

## 1. Introduction

According to the American Cancer Society, bladder cancer is the eighth most common cancer in the United States, with an estimated 82,000 new cases and 16,700 deaths in 2023 [1,2]. Of these cases, 75% were non-muscle invasive bladder cancer (NMIBC) [2]. The American Urological Association (AUA) stratifies NMIBC into low-, intermediate-, and high-risk based on histologic grade, tumor staging, number of tumors, presence of recurrence, Bacillus Calmette–Guérin (BCG) responsiveness, presence of variant histologies, and presence of lymphovascular or prostatic urethra invasion [2]. The European Association of Urology (EAU) guidelines use a similar risk-stratification system; however, in contrast to the AUA guidelines, the EAU system incorporates age > 70 as a risk factor and also includes a very high-risk category. The very high-risk category includes patients who not only have high-grade disease and/or CIS but also present additional risk factors, including age > 70, multifocal papillary tumors, and >3 cm [3]. With low-risk NMIBC, despite very low rates of progression, the risk of recurrence is approximately 40–50% within 5 years [2,4]. For high-risk diseases, even though up to 80% will have an initial response to intravesical BCG, the overall rate of high-grade recurrence or progression is approximately 50% [5,6]. For T1 disease, especially when present on a repeat resection, progression rates approach 80% [2]. For cases that do not respond to initial intravesical therapy, secondary bladder-sparing therapies have a success rate in the 20–40% range [5,6,7]. Radical cystectomy is the gold standard and has excellent outcomes when initiated in this setting, but many patients prefer bladder-sparing approaches. Thus, there is a significant impetus to develop novel therapies and delivery of existing therapies to decrease recurrence, progression, and need for radical therapy in the NMIBC population. Additionally, bladder cancer is one of the costliest malignancies in the healthcare system [8]. With the high recurrence rates and strict surveillance schedules, NMIBC contributes significantly to the economic burden of healthcare in the United States [9]. Given the national shortage of the most-studied intravesical therapy (BCG), the financial burden of a high recurrence rate, and non-adherence to standard therapy, this review aims to highlight recent publications and current clinical trials within the United States that investigate novel treatment sequences, therapeutic agents, and dosing strategies and place the ongoing clinical trials into the context of published literature on these emerging treatment strategies for NMIBC.

## 2. Methods

The current active clinical trials for NMIBC with therapeutic intent were queried by the National Library of Medicine (NLM) (www.clinicaltrials.gov, accessed on 18 March 2024) between the years 2015 and 2024. The search criteria on the NLM site included ‘Non-muscle Invasive Bladder Cancer \(NMIBC\)’, ‘United States’, ‘adult (18–64)’, ‘older adult (65+)’, ‘phase 2, and 3’ and ‘interventional’. The trials chosen were then stratified based on low-, intermediate-, or high-risk NMIBC risk. The high-risk NMIBC trials chosen were additionally classified based on their BCG treatment into BCG-naïve and BCG-unresponsive. Trials including both BCG-naïve and BCG-unresponsive patients were included in both categories. Trials with an active status (recruiting, active but not recruiting, ongoing) were selected, while trials with a status of complete, terminated, suspended, or withdrawn were excluded. The search for literature on current NMIBC therapeutics was queried from PubMed using the keywords ‘non-muscle invasive bladder cancer’, ‘low-risk NMIBC’, ‘high-risk NMIBC’, ‘intermediate-risk NMIBC’, ‘BCG-naïve’, and ‘BCG-unresponsive’. The papers selected were published in the English language and primarily published within the past 5 years. To perform this review, first, the entire study team, including translational researchers focusing on bladder cancer, a fellowship-trained urologic oncologist, and a urologist in training, agreed upon the search criteria. S.M. and K.H. then extracted and reviewed studies meeting the above criteria. The content of the studies and suitability for inclusion were then reviewed by the entire study team for final inclusion.

## 3. Results and Discussion

### 3.1. Low-Risk Non-Muscle Invasive Bladder Cancer

The current standard of care for low-risk NMIBC is TURBT with a single dose of intravesical chemotherapy (gemcitabine or mitomycin C) in patients with presumed or known low- to intermediate-risk bladder cancer unless there is concern for perforation or extensive resection, according to the AUA and the National Comprehensive Cancer Network Guidelines [2,10]. If a visually complete resection is performed in low-risk cancer, then repeat TURBT is not recommended. Induction intravesical therapy is not indicated in these patients, and following initial management, patients are placed on a surveillance schedule [2,10]. While low-risk NMIBC has a low rate of progression, it has a considerable recurrence rate—cited as high as 40–67% [4,11], illustrating the need to explore new therapies even in the low-risk space. 

One issue with the high recurrence rate is incomplete adherence amongst urologists. A 2020 review analyzing six studies, including 1193 patients, found that the overall compliance for one-time post-TURBT intravesical chemotherapy instillation was 53% [12]. Within this review, the use of intravesical chemotherapy was reported to be as low as 27–50% amongst surgeons surveyed from five Urological Surgery Quality Collaborative practices in the US, even in patients noted to be “ideal cases” [13]. Barriers to use that have been historically cited include concerns about side effects, logistical and workflow difficulty (not ordered pre-operatively, appropriate resources not available in the post-anesthesia care unit, uncertainty about what phase of care the instillation is performed and who is responsible for what task, etc.), and lack of training, among others [12,14,15]. In response to the limited use of post-operative intravesical chemotherapy despite the recommendation, the clinical trial SWOG S0337 sought and successfully demonstrated further evidence of the benefit of intravesical chemotherapy with the use of gemcitabine. This trial showed that patients with low-risk NMIBC who received a single dose post-operative instillation of gemcitabine compared with saline for a 60-min dwell time had an absolute reduction in recurrence of 12% at 4 years. Prior to this, it had been more than two decades since an intravesical agent had demonstrated promising results as a single post-operative dose, and unlike agents such as mitomycin C, which have been shown to have toxic effects, gemcitabine did not demonstrate evidence of side effects beyond those of the TURBT alone [16].

The current intravesical chemotherapy agents used typically require a 60-min dwell time in the bladder. By shortening the dwell time while still maintaining the efficacy of the treatment, this could benefit both surgeons and medical staff and possibly combat some barriers to the use of single post-TURBT instillation. A Japanese study in 2021 evaluated the efficacy of a single short retention instillation of pirarubicin following TURBT and compared it to TURBT alone in patients with low-risk NMIBC. The results showed that 15-min retention of pirarubicin immediately post-operatively resulted in higher one- and five-year recurrence-free survival rates. This study also demonstrated that in various bladder cancer cell lines, pirarubicin exerted a similar antiproliferative effect at low concentrations at 15 min and 60 min, while mitomycin C, a commonly used intravesical chemotherapy agent, was more effective at 60 min compared with 15 min. Pirarubicin is not currently approved for use for bladder cancer in the United States. The study was also limited by a small sample size and the lack of a comparator arm for a longer dwell time/standard chemotherapy, such as gemcitabine. This could encourage further research into pirarubicin or other novel effective intravesical chemotherapeutic agents that can be retained for shorter periods of time, as it could increase the utilization of the recommended single-dose TURBT instillation [17]. However, the current evidence supports a 60-min dwell time as the most efficacious given the higher level of evidence.

Given the disparity in the use of single-instillation intravesical chemotherapy among urologists in the United States and Europe, as well as the low, but still present, risk of adverse effects of intravesical chemotherapy, Onishi et al. performed a randomized study published in 2016 that compared adjuvant therapy with continuous saline bladder irrigation to single immediate instillation of mitomycin C in patients with low- to intermediate-risk NMIBC. The results demonstrated no significant difference in recurrence-free survival at one, three, and five years post-TURBT [18]. Note that patients who received continuous saline bladder irrigation underwent 15 h of irrigation. As mentioned above, later studies that evaluated the efficacy of single-dose intravesical gemcitabine compared it to intravesical instillation of saline with a 60-min dwell time and found the chemotherapy agent to be superior [16]. To our knowledge, continuous bladder irrigation has not been directly compared to intravesical gemcitabine. TURBT with a single instillation of intravesical chemotherapy is typically an outpatient procedure; however, the use of continuous bladder irrigation for the designated amount of time from this study would require an overnight hospitalization, which would both present an added healthcare expense and limit the facilities where this procedure could be performed. That being said, with the variable utilization of single intravesical chemotherapy in low-risk NMIBC, continuous bladder irrigation does appear to be an option in situations whereby an inpatient stay is feasible and intravesical chemotherapy is not used for reasons such as lack of resources, concern for side effects, patient refusal, etc.

In addition to evaluating different options for adjuvant therapy for low-risk NMIBC, there is also the opportunity to review novel methods for surgical resection of these tumors. Li et al. compared standard TURBT to two novel resection methods: en-bloc resection with a pin-shaped electrode (pin-ERBT) and transurethral holmium laser resection of bladder tumor (HoLRBT) in 115 patients. Amongst the patients in the three groups, there was no statistically significant difference in sex, age, smoking history, lesion size or location, or pathological grade. All patients had a primary, solitary Ta bladder tumor with a diameter of <3 cm, stratifying them to the low-risk category. The TURBT and HoLBRT groups had recurrence rates of 38% and 40%, respectively, at 24 months; however, the recurrence rate in the pin-EBRT group was significantly lower at 10%. It is thought that contributing factors to the lower recurrence rate in the pin-EBRT group include precision of resection due to the thin, flexible electrode, the ability to set the cutting range prior to resection and partial blockage of blood supply provided with the electrode, which can decrease the risk of recurrence and metastasis secondary to blood-borne dissemination, and the ability to remove the specimen intact, thus decreasing the risk of recurrence due to implantation [19]. Further evidence regarding en-bloc resection includes Paciotti et al., who evaluated long-term oncologic outcomes in patients with NMIBC who were treated with en-bloc transurethral resection (EBRT). Among their 74-patient cohort, including low-, intermediate- and high-risk patients, in a five-year follow-up period, none of their patients who received EBRT progressed to muscle-invasive bladder cancer, and 77% of their entire cohort was recurrence-free. In the subset with low- and intermediate-risk NMIBC, they reported an 81% 5-year recurrence-free rate [20]. While these data are limited by potential selection bias and the lack of a comparator arm, the signal from these studies suggests the need for randomized comparisons to evaluate the ideal resection technique. In comparison to high-risk diseases, there are fewer ongoing trials focusing on low- and intermediate-risk diseases. For low-risk diseases, where recurrence is much more common than progression, recurrence is the primary outcome of studies rather than progression. The ENVISION trial (NCT05243550) is assessing the complete response rate, as well as the duration of complete response, the durability of complete response, disease-free survival, and the rate of adverse events, in patients with low-grade intermediate-risk NMIBC who are treated with UGN-102 in a phase III, single-arm, multicenter study [21]. UGN-102 is a sustained-release, hydrogel-based formulation of mitomycin designed as an intravesical therapy that is thought to provide longer exposure to the bladder wall and is thus being investigated as an alternative to TURBT. Early data from the ATLAS trial (NCT04688931), comparing UGN-102 to standard TURBT, demonstrated a comparable complete response rate at 3 months between the two at 65% for UGN-102 and 64% for TURBT. Preliminary data for ENVISION show an even better complete response rate at 79%, with data regarding duration and durability of response, disease-free survival, and adverse effects still in progress [22]. If this ongoing study shows promising results with regards to recurrence rates in low-grade intermediate-risk NMIBC, then it could be further considered for patients with low-risk NMIBC and potentially provide an alternate option to TURBT, subsequently saving patients from repeat visits to the operating room and the associated risks of anesthesia or discomfort of in-office fulguration.

An ongoing single-arm, phase II window of opportunity study, J18158 (NCT03914794), is currently being conducted in 4-5 high-volume bladder cancer centers in the United States. Patients with recurrent low- or intermediate-risk NMIBC are treated with pemigatinib for 4–6 weeks prior to standard TURBT. Pemigatinib is an oral inhibitor of fibroblast growth factor receptors (FGFRs) 1, 2, and 3 that is currently FDA-approved for advanced and metastatic cholangiocarcinoma and myeloid/lymphoid neoplasms with certain FGFR rearrangements [23,24]. In this study, the response rate of pemigatinib therapy, adverse effects, response rate stratified by FGFR3 mutational status, risk group, the concentration of the drug in urothelial tissue at the time of TURBT, and relapse-free survival serve as primary and secondary outcome measures [25]. Data from these trials aim to guide further investigation into pemigatinib or other FGFR inhibitors for the treatment of NMIBC.

While the aim of this review was to discuss treatment modalities, it is important to note that there is also ongoing advancement in NMIBC surveillance for the early detection of recurrence and decreasing the burden of surveillance. Frequent cystoscopies required for surveillance can be burdensome and uncomfortable for patients and are also a source of expense for the healthcare system. The use of urinary biomarkers for surveillance is less invasive and less time-consuming for patients and would also likely be more cost-effective. Further development of urinary biomarkers could have significant benefits since decreasing cystoscopy discomfort has been shown to be a research priority, as demonstrated by the Bladder Cancer Advocacy Network: Patient Survey Network [26]. It is important that novel biomarkers can demonstrate a high negative predictive value to be sufficiently useful to avoid cystoscopy, which has a negative predictive value of 95% [27]. There are currently urinary biomarker tests approved by the US Food and Drug Administration; however, due to low specificity and sensitivity, they have limited use in a clinical setting [28]. Further research is ongoing, including urine tumor DNA and urine mRNA, which may eventually be suitable for use in clinical practice. 

### 3.2. Intermediate- and High-Risk Non-Muscle Invasive Bladder Cancer

As with low-risk NMIBC, the initial treatment for diagnosis, staging, and tumor clearance is TURBT for intermediate- and high-risk diseases. Unlike low-risk NMIBC, a repeat resection to ensure endoscopic clearance and for additional staging information is recommended for high-grade disease, especially high-grade T1 [2]. Repeat resection provides additional staging information and has been shown to improve response to intravesical therapy by improving disease clearance [29]. The standard of care for intermediate-risk patients is the option of a 6-week induction course of intravesical chemotherapy or BCG, with the option to continue with maintenance therapy for either if there is a complete response. Patients with high-risk NMIBC are recommended to have 6 weeks of induction therapy with BCG followed by 3 years of maintenance BCG if they have a complete response, pending BCG availability and patient tolerance [2,10]. BCG status plays a significant role in patients who require further treatment, and below, we discuss novel treatment options for intermediate- and high-risk NMIBC based on BCG status, with ongoing clinical trials outlined in Table 1 and Table 2. Trials are also comparing novel treatments to BCG as the initial therapy for high-risk diseases. We describe these emerging therapeutic strategies below.

### 3.3. BCG-Naïve

Since BCG was identified as having efficacy for preventing both recurrence and progression in NMIBC in 1976, it has remained the gold standard [30]. The initial response rate is 70%, a high standard [31]. However, despite this high initial response rate, recurrence and progression occurs in approximately 40% of patients [7]. Additionally, since 2019, there has been a critical BCG shortage due to a single manufacturer of the strain used in the US, which has provided an ongoing challenge in the treatment of NMIBC. These factors have all spurred increased development of novel therapies. In response, a variety of treatment options have been, and are currently, under investigation for patients with BCG-naïve NMIBC.

Researchers have investigated altered BCG treatment schedules as a response to the ongoing BCG shortage. In 2021, an observational study out of Japan compared recurrence risk, risk of progression, and cancer-specific death in a large cohort of patients who were assigned to receive varying amounts of BCG treatments. Specifically, Miyake et al. discussed recurrence-free survival in patients who underwent six doses of induction BCG plus maintenance BCG compared to patients who received seven or eight doses of induction BCG alone. All patients were BCG-naïve and were risk stratified into the European Association of Urology high- or highest-risk categories. In non-landmark analysis, there was a higher rate of recurrence-free survival amongst the patients who received six doses of induction BCG plus maintenance BCG compared with patients who only received seven or eight doses of induction BCG; however, when utilizing landmark analysis, there was no significant difference in recurrence-free survival between the two groups at 12, 18, or 24 months [32]. More recently, a phase II clinical trial at the Memorial Sloan Kettering Cancer Center in New York released 5-year follow-up data supporting the feasibility of a modified BCG schedule. They enrolled 76 patients who underwent two induction BCG courses (for a total of 12 intravesical instillations) after visibly complete TURBT. Among those patients, 78% were BCG-naïve. Time was assessed from a 7.5-month landmark, and among BCG-naïve patients, recurrence-free survival was 65% compared with 82% for patients with prior BCG; however, this difference was not statistically significant [33]. The recurrence-free survival rate is comparable to the SWOG-8507 trial, which showed a 60% recurrence-free survival at 5 years among patients who received induction BCG followed by 3 years of maintenance BCG [34]. Ultimately, this suggests that similar outcomes can be obtained regarding recurrence rates requiring significantly fewer vials of BCG. In addition to preserving vials of BCG, the use of different formulations could help rectify the current BCG shortage. A clinical trial in its early stages, the EVER trial (NCT05037279), is investigating the efficacy of the VERITY-BCG strain. This randomized, double-blind clinical trial seeks to compare two different BCG formulations, VERITY-BCG and OncoTICE (standard BCG), in BCG-naïve patients with NMIBC. The study aims to show non-inferior results regarding two-year recurrence-free survival between the two BCG formulations [35].

One alternative to BCG that began being investigated in response to the shortage is a combination: sequential intravesical therapy with gemcitabine followed by docetaxel. Treatment with gemcitabine and docetaxel had previously been shown to be an effective treatment for BCG-unresponsive NMIBC [7,36]. More recently, McElree et al. compared the recurrence-free survival and adverse events of sequential intravesical gemcitabine and docetaxel versus BCG in patients with BCG-naïve high-risk NMIBC. In this retrospective cohort study, 312 patients were included from 1 January 2011 to 31 December 2021. There was no statistically significant difference in tumor pathology between the gemcitabine/docetaxel groups and the BCG group, with both groups having the highest number of patients with T1 tumors. Additionally, there was no statistically significant difference in tumor size, the number of patients with multifocal tumors, or the number of patients with pre-treatment CIS-containing tumors. Treatment with gemcitabine and docetaxel for BCG-naïve high-risk NMIBC was found to have higher recurrence-free survival than treatment with BCG [37]. These data supported the use of gemcitabine and docetaxel in high-risk BCG-naïve NMIBC while awaiting results from ongoing clinical trials. In a small cohort of patients with primarily high-risk NMIBC, among which 85% were BCG-naïve (28/33), Zeng et al. demonstrated that the use of intravesical gemcitabine as a first-line therapy in the BCG shortage era resulted in a roughly 85% complete response rate and, in those patients, an approximately 87% 6-month recurrence-free survival and 76% 12-month recurrence-free survival. This patient cohort was primarily made up of patients with T1 tumors, and 91% of the patients fell in the AUA high-risk category [38]. While this study utilized a small patient cohort with a short follow-up interval, it provides more data in support of an intravesical chemotherapy agent in BCG-naïve patients. Clinical trials comparing treatment with gemcitabine plus docetaxel and intravesical BCG in BCG-naïve patients are currently being performed. The BRIDGE trial (NCT05538663) is a phase IIIb clinical trial comparing the event-free survival of BCG-naïve patients with NMIBC treated with BGC versus gemcitabine/docetaxel. Secondary objectives in this trial include cystectomy-free survival, cancer-free survival, safety, toxicity, and cancer-specific and bladder cancer-specific quality of life changes from baseline to treatment [39,40]. The GEMDOCE (NCT04386746) trial is an active single-arm phase II study that also focuses on gemcitabine and docetaxel in BCG-naïve patients. This study seeks to provide efficacy and safety data on this treatment regimen in BCG-naïve patients with intermediate- and high-risk NMIBC [41].

An area of current interest is the use of PD-(L)1 inhibitors, either as monotherapy or an adjuvant treatment with BCG. PD-1 is typically expressed by T-cells. When a PD-1 receptor is bound by its ligand (PD-L1), it down-regulates the T cell response, and in cancer, this can help tumor cells evade the immune response. Inhibitors to PD-1 or PD-L1 can help prevent this interaction [42,43]. CREST (NCT04165317) is a phase III clinical trial evaluating the efficacy and safety of the PD-1 inhibitor sasanlimab plus BCG in comparison to BCG alone in BCG-naïve high-risk NMIBC. The primary endpoint is event-free survival, with the hypothesis that sasanlimab plus BCG therapy is superior to BCG monotherapy [44,45]. The PATAPSCO (NCT05943106) study is also investigating a monoclonal antibody that interacts with the PD-(L)1 pathway. Durvalumab is a PD-L1 inhibitor that has demonstrated effectiveness in advanced urothelial carcinoma and is now being investigated for a role as a treatment modality for NMIBC [46]. A phase IIIb, single-arm, multicenter US study is currently underway investigating the safety, tolerability, and efficacy of durvalumab plus BCG in high-risk NMIBC in patients who have not received prior systemic therapy for bladder cancer and are BCG-naïve. Endpoints focus on adverse events, patient tolerability, response rate, survival, and quality of life [47]. Pembrolizumab is another PD-1 inhibitor that has been approved by the US FDA for the treatment of many cancers, including BCG-unresponsive NMIBC in patients who were ineligible for or declined cystectomy [48]. Further studies are now underway to assess the effectiveness of pembrolizumab in BCG-naïve patients with NMIBC. One cohort of the KEYNOTE-676 (NCT03711032) trial is looking to compare the event-free survival of up to 5 years of BCG-naïve patients with high-risk NMIBC who receive pembrolizumab plus induction and a full maintenance course of BCG, who receive pembrolizumab plus induction and a reduced maintenance course of BCG, and those who receive BCG monotherapy with standard induction and maintenance therapy. Similarly, there is a phase II clinical trial underway in which one cohort assesses the rate of high-grade recurrence at 6 months in BCG-naïve patients with high-risk T1 NMIBC who receive pembrolizumab in combination with standard induction and maintenance BCG [49]. As these studies come to a close and study data become available, immune checkpoint inhibitors may find a place in the treatment algorithm in BCG-naïve NMBIC.

The SunRISe trials are investigating an intravesical drug delivery product that provides the continuous release of gemcitabine into the bladder, known as TAR-200. There are multiple SunRISe trials currently underway. SunRISe-3 (NCT05714202) is specifically focused on BCG-naïve patients with high-risk NMIBC. In this phase III study, event-free survival in patients who receive TAR-200 plus the PD-1 inhibitor cetrelimab or TAR-200 alone is being compared to BCG alone [50,51]. SunRISe-1 and 5 will be discussed with other ongoing clinical trials for BCG-unresponsive NMIBC.

Gene therapy is another area of investigation for novel NMIBC treatments. LEGEND (NCT04752722) is a phase I/II trial evaluating the efficacy and safety of EG-70. This is a novel non-viral gene therapy that can elicit a local immune response when delivered intravesically. Phase I of this study will assess safety and recommended dose. For phase II, BCG-naïve patients with NMIBC will be enrolled in a single-arm cohort (with BCG-unresponsive patients in a separate cohort) and treated with the recommended dose from phase I. Outcomes, including complete response rate at various time points, type, incidence and severity of emergent adverse events, progression-free survival, and duration of response, will be measured in each separate cohort [52]. 

The PIVOT-006 trial (NCT06111235) recently dosed their first patient in early 2024. This trial is investigating cretostimogene grenadenorepvec (CG0070), a conditionally replicating oncolytic adenovirus that preferentially replicates within and kills Rb-defective cancer cells [53], following TURBT in patients with intermediate-risk bladder cancer. With the BCG shortage, BCG is often reserved for high-risk cancer, thus creating a difficult scenario for those diagnosed with intermediate-risk NMIBC. Therefore, the investigators are hoping to demonstrate CG0070 as an efficacious oncolytic immunotherapy agent for this patient population [54]. This is a phase III, open-label, randomized clinical trial. To be enrolled, patients must have pathologically confirmed intermediate-risk NMIBC with all visible diseases removed using TURBT within 12 weeks of study randomization. Patients are randomized into two arms: arm A, which will be treated with CG0070 and n-dodecyl-b-d-maltoside (DDM), a transduction-enhancing agent, after TURBT, and arm B, which will undergo observation after TURBT. Outcomes measured include recurrence-free survival and incidence of adverse events. The estimated study completion is in 2030 [55]. CG0070 is also being investigated as a therapy option in BCG-unresponsive patients, and those trials will be further discussed below.

### 3.4. BCG-Unresponsive

For the purposes of this review, we use the term BCG-unresponsive, which includes patients with BCG-refractory and BCG-relapsing cancer. Currently, guidelines recommend radical cystectomy in BCG-unresponsive patients who are appropriate surgical candidates and are willing to undergo cystectomy [2,10]. Despite the favorable quality-of-life outcomes after radical cystectomy [56], many patients desire a bladder-sparing option in this circumstance. A study in which patients with BCG-unresponsive NMIBC were asked to make hypothetical choices regarding treatment options showed that to increase the time until radical cystectomy from 1 to 6 years, patients were willing to accept an almost 44% increased risk of cancer progression and approximately 66% increased risk of serious side effects [57]. A study of United States practice patterns showed that only 25% of surveyed urologists would consider radical cystectomy for BCG-unresponsive high-grade papillary disease. Meanwhile, more than half regularly utilized intravesical chemotherapy for these patients [58]. This has spurred the use of novel combinations of existing intravesical therapies, intravesical gene therapies, intravenous immunotherapy, and novel methods of drug delivery in BCG-unresponsive NMIBC. Sequential gemcitabine and docetaxel have been increasingly studied and used in clinical practice for BCG-unresponsive disease in comparison to previously used regimens, such as gemcitabine monotherapy, valrubicin, and mitomycin. In 2015, a single institution retrospective study first reported on the use of this combination therapy in patients exposed to prior BCG. This regimen, which has now become standard, includes 1 g of gemcitabine instilled for 90 min, which is then removed and followed by 37.5 mg of docetaxel instilled for 90 to 120 min. Induction therapy is a 6-week course, followed thereafter by monthly maintenance therapy for up to 24 months. With this regimen, Steinberg and colleagues reported an overall 66% treatment success at first surveillance (median 5.9 months), 54% at one year, and 34% at 2 years. When stratifying treatment success by a number of prior BCG failures, there was no statistically significant difference, and within their patient cohort, more than 80% had BCG-unresponsive NMIBC [7]. While these results have been seen as promising, it is important to note that these were not prospectively monitored outcomes. In a larger multi-institutional retrospective cohort of BCG-unresponsive NMIBC, sequential gemcitabine/docetaxel reported a 46% two-year recurrence-free survival with a 96% cancer-specific survival and a low two-year progression rate at 7% [36]. The same limitations apply to the lack of a comparator arm and retrospective nature. Most recently, Chevuru et al. reported on the long-term outcomes of 97 patients treated with gemcitabine/docetaxel for high-risk NMIBC after BCG failure. Of this cohort, 35% were BCG-unresponsive. They found high-grade recurrence-free survival to be 60% at one year, 50% at two years, and 30% at five years. These rates were similar in the subset that was categorized as BCG-unresponsive (67%, 50%, and 28% at one, two, and five years, respectively, in the BCG-unresponsive group). Additionally, the 5-year bladder preservation rate was 75% in the overall cohort and 74% amongst the BCG-unresponsive group, and the 5-year cancer-specific survival was 91% amongst the entire cohort and 92% in the BCG-unresponsive subset of patients [59]. In summary, there is retrospective evidence that sequential gemcitabine/docetaxel has superior outcomes to those reported in the literature for single-agent therapy. Prior studies have demonstrated recurrence-free survival in BCG-unresponsive or intolerant patients who received monotherapy with intravesical gemcitabine as 33% at 3 years [60] and a 21% complete response rate among patients with BCG-unresponsive CIS who were treated with intravesical valrubicin monotherapy [61]. It must be acknowledged that a preponderance of the evidence was published from a single center, and this may not be fully generalizable. There is also the potential for a selection bias where high volume disease, more extensive CIS, and more of the T1 population elects for immediate radical cystectomy, and thus, it can be problematic to extrapolate to all BCG-unresponsive cases without standardized selection criteria that would be offered in a prospective study.

There has also been a longstanding effort to incorporate intravesical gene therapy as an option for BCG-unresponsive disease. Given the ability to directly instill into the bladder using a catheter, allowing for direct contact between a gene therapy vector and the diseased cells, bladder cancer is an ideal target for gene therapy. In December 2022, nadofaragene firadenovec became the first gene therapy approved for genitourinary cancer by the US FDA [62]. A phase III, multicenter, single-arm study enrolled patients with BCG-unresponsive NMIBC at 33 sites in the United States and evaluated outcomes after intravesical therapy with nadofaragene firadenovec. Patients recruited were at least 18 years old with BCG-unresponsive NMIC and an Eastern Cooperative Oncology Group status of two or fewer. Any patients with evidence of upper tract disease, carcinoma within the prostatic urethra, lymphovascular invasion, micropapillary disease, or hydronephrosis were excluded. There were 157 patients ultimately enrolled, with staging as follows: 52% CIS only, 22% Ta, 13% Ta plus CIS, 10% T1, and 3% T1 plus CIS. Patients were divided into a CIS cohort (patients with CIS ± high-grade Ta or T1 disease) and a high-grade Ta or T1 cohort (patients with high-grade Ta or T1 disease with no CIS). The primary endpoint was the proportion of patients in the CIS cohort with a complete response at any time within 12 months after their first dose of nadofaragene firadenovec. Secondary endpoints included durability of complete response, high-grade recurrence-free survival in the Ta or T1 cohort, radical cystectomy-free survival in both cohorts, overall survival in both cohorts and safety in all patients. In the CIS cohort, 53% of patients had a complete response, and ultimately, all complete responses were documented by month three. Amongst all patients, 60% had a complete response at 3 months, with a median duration of complete response of 7.3 months. At 6, 9, and 12 months, 48%, 42%, and 30% of patients, respectively, remained free from high-grade recurrence. The therapy was overall well tolerated. While 70% of all patients had drug-related adverse effects, a majority were classified as grade 1 or 2 and were transient. The most common adverse event was discharge around the catheter during instillation (25% of patients). Grade 3 or 4 adverse events were noted in 18% of patients, including urgency, bladder spasms, urinary incontinence, syncope, and hypertension. Three patients opted to discontinue the study due to adverse effects. Additionally, the treatment schedule of nadofaragene firadenovec is one instillation every three months, which is less burdensome than many intravesical therapies [5]. The approval of this treatment provides an additional bladder-sparing option for those patients who decline or are unfit for radical cystectomy and serves as a stepping stone for the development of other similar therapies.

As with BCG-naïve disease, where PD-(L)1 agents are being investigated, the activity of these agents in more advanced bladder cancer has led to an investigation in the localized setting. The KEYNOTE-057 (NCT02625961) study resulted in the FDA approval of intravenous pembrolizumab for the treatment of BCG-unresponsive carcinoma in situ. KEYNOTE-057 was a phase II randomized parallel study in which three cohorts of which Cohorts A and B received pembrolizumab and Cohort C received either pembrolizumab/vibostolimab coformulation or favezelimab/pembrolizumab coformulation. Pembrolizumab is an intravenous infusion administered at 200 mg every 3 weeks. Thus far, the results show a complete response rate of 41%, with the duration of response being 16.2 months for those 41% of patients. Among the patients who had a complete response (41% of the cohort), 46% maintained their complete response for 12 months or longer (representing 19% of total starting therapy). Overall survival was 91% at 36 months. Treatment-related adverse events were documented in 66% of patients, with 13% of patients experiencing grade 3 or 4 adverse events. The most common overall adverse events were diarrhea and fatigue (11% of patients each), with the most common grade 3 or 4 adverse events being hyponatremia and arthralgias (2% of patients each). In KEYNOTE-057, patients continued treatment until unacceptable toxicity, persistent or recurrent high-risk NMIBC or progressive disease, or up to 24 months of therapy without disease progression. The median duration of treatment was 4.2 months [6], notably less than the maximum of 24 months possible. Pembrolizumab presents a bladder-sparing treatment option with a reasonable treatment schedule that avoids the specific side effects of intravesical therapies. With its FDA approval, its use is increasing in the clinical setting and will hopefully continue to serve as an efficacious option for select patients [63,64]. The PD-1 inhibitor sasanlimab, discussed above as its efficacy when combined with BCG, is being investigated in the BCG-naïve population and is also being studied as a single agent therapy in BCG-unresponsive NMIBC. Cohort B of the CREST trial (NCT04165317) included patients with BCG-unresponsive, high-risk NMIBC that had a CIS recurrence within 12 months of completion of adequate BCG therapy or a Ta or T1 recurrence within 6 months of completion of BCG. These patients will be treated with sasanlimab alone, and primary outcome measures include a complete response rate and event-free survival. Study completion is expected in December 2026 [45].

An additional clinical trial investigating the use of pembrolizumab is CORE001 (NCT04387461). CORE001 is a single-arm, phase II trial that looks at the complete response rate with the usage of a combination of CG0070 and pembrolizumab within 35 patients. As mentioned above, CG0070 is a conditionally replicating oncolytic adenovirus that preferentially replicates within and kills Rb-defective cancer cells [53]. Treatment with the CG0070 and pembrolizumab combination led to an overall complete response rate of 85% [65]. A study that focuses solely on CG0070 in BCG-unresponsive patients is BOND003 (NCT04452591). BOND003 is a phase III single-arm trial focusing on BCG-unresponsive patients with persistent or recurrent CIS with or without high-grade Ta or T1 disease. In this study, patients are treated with CG0070 and a transduction-enhancing agent, n-dodecyl-b-d-maltoside. The complete response rate at any time point for these patients was 76%. Further, 74% of the patients who achieved complete responses had their responses last for a minimum of 6 months. Notably, early data also demonstrated that 31% of patients who did not respond to their first course of treatment were salvaged with re-induction. Most adverse events experienced by patients were of grade 1 or 2, with the most common being bladder spasms and pollakiuria occurring in 20% and 16% of the patients in the safety population, respectively. No patients experienced grade 3 or higher adverse events. Due to these promising early results, the FDA granted CG0070 a breakthrough therapy designation and a fast-track designation for the treatment of patients with BCG-unresponsive CIS (±papillary Ta or T1 disease) in late 2023 [66].

Small molecule inhibitors are a form of targeted cancer treatment whose mechanism is to inhibit the target proteins’ function [67]. One instance we see of a small molecule inhibitor is with the drug erdafitinib. Erdafitinib is an ATP-competitive inhibitor of FGFR 1–4 [68]. Erdafitinib has demonstrated clinical activity in participants with solid tumors with alterations in the FGFR pathway. THOR-002 (NCT04172675) is a study that focuses on the efficacy of erdafitinib. THOR-002 is a multicenter, phase II randomized clinical trial in BCG-unresponsive patients with high-risk NMIBC and specific FGFR mutations or fusions investigating treatment with erdafitinib. Patients treated with erdafitinib received the medication orally once every 28 days until two years of treatment were completed, there was disease recurrence, patients experienced intolerable toxicity from the treatment, consent was withdrawn, the investigator decided to discontinue treatment, or the study was terminated. These patients were compared to patients treated once weekly with intravesical gemcitabine or mitomycin for four induction doses, followed by at least six months of monthly maintenance therapy. Recurrence-free survival was the primary outcome, with secondary outcomes including time to progression, overall survival, plasma concentration of erdafitinib, and number of participants with adverse events. There was evidence that erdafitinib reduced the risk of recurrence of disease or death by 72% compared with intravesical chemotherapy in data from cohort 1 in this clinical trial. Cohort 1 included 49 patients randomized to receive erdafitinib and 24 patients randomized to receive intravesical chemotherapy. Among the patients who received erdafitinib, grade 3 or 4 treatment-related adverse events occurred in 31% [69,70]. Recurrence rates are promising with this treatment modality; however, it is selective for patients who specifically have certain FGFR mutations of fusions, limiting the number of patients who may benefit. Also, while it is an oral treatment, which may be more appealing to certain patients than an intravesical therapy, initial data show a high rate of treatment-related adverse events. This study is estimated to be completed in March 2025 [71].

As discussed in the BCG-naïve setting, TAR-200 is an intravesical drug delivery product that provides continuous release of gemcitabine into the bladder after being inserted by catheter and exchanged every 3 weeks during the 24-week induction course. Currently, there are two trials that investigate the efficacy of TAR-200 in BCG-unresponsive patients, SunRISe-1 (NCT04640623) and SunRISe-5 (NCT06211764). SunRISe-1 is a phase II randomized trial that focuses on evaluating the efficacy of TAR-200 and cetrelimab using overall clinical response and disease-free survival. This study began in December 2020 and released interim results in April 2023, in which TAR-200 was found to have a complete response rate of 73% of patients in the TAR-200 arm compared with the complete response rate of 38% in the cetrelimab arm. A complete response is defined as the absence of a high-grade disease on cystoscopy and urine cytology at any time point. At the median follow-up of 10.6 months, 15 of the 16 patients in the TAR-200 arm who achieved complete response had an ongoing response, with six of the patients who achieved complete response continuing their response beyond 12 months. At the time of preliminary data’s release, none of the TAR-200 arm patients who achieved complete response had experienced recurrence or progression. The estimated study completion is July 2027 [72,73]. Updated interim data from this study were released in October 2023, demonstrating a 77% complete response rate in patients with high-risk NMIBC with CIS treated with TAR-200 monotherapy, with 22% of patients experiencing treatment-related adverse events, a majority of which were of grade 1 or 2. In response to these results, the FDA has granted a breakthrough therapy designation to TAR-200 for the treatment of patients with BCG-unresponsive, high-risk NMIBC who cannot or opt not to undergo cystectomy [74]. SunRISe-5 is a phase II randomized study comparing TAR-200 to mitomycin C or gemcitabine in NMIBC patients within 1 year of the last dose of BCG. This study is estimated to start on 26 April 2024 [75]. The SunRISe trials allow for a broad scope of what TAR-200 is capable of, its efficacy, and how TAR-200 interacts across a variety of disease settings.

Additional immune-modulating therapies include N-803, an interleukin-15 superagonist specifically set to increase natural killer cell and CD8+ T-cell persistence [76]. QUILT-3.032 (NCT03022825) is a phase II/III, single-arm clinical trial that treats patients with either intravesical N-803 plus BCG or intravesical N-803 monotherapy. Patients included in this study must have BCG-unresponsive, histologically confirmed high-grade NMIBC [77]. This study is ongoing; however, at the time of preliminary data release, which evaluated patients who received N-803 plus BCG, 160 patients had been enrolled, 83 with CIS and 77 with papillary disease. Patients were divided into two groups, including patients with CIS and those with papillary disease, for data analysis. Patients with CIS showed a complete response rate of 71% with a median duration of response of 24.1 months, and 91% avoided cystectomy. Patients with papillary disease had a 57% 12-month disease-free survival (DFS) rate, 48% had a 24-month DFS rate, and 95% avoided cystectomy. In the entire patient cohort, there was a 99% bladder cancer-specific survival rate at two years, and there were no grade 4 or 5 treatment-related adverse events [76]. 

Lastly, TARA-200 is an investigational cellular therapy made to target Toll-Like Receptor 4 in NMIBC. Currently, TARA-200 is being utilized in the ADVANCED2 trial (NCT05951179). ADVANCED2 is a phase Ib/II, single-arm clinical trial testing the efficacy of TARA-200 in two cohorts, BCG-naïve and BCG-unresponsive. ADVANCED2 comes after the preliminary findings of the ADVANCED1 (NCT05085977) trial and seeks to find a proper dosage for TARA-200 in both BCG-naïve and BCG-unresponsive patients. Of the 102 patients enrolled in ADVANCED2, 75 are currently categorized as BCG-unresponsive. For those 75 patients, this study aims to find the incidence of high-grade CR after treatment with TARA-002 within 3 to 24 months [78].

## 4. Conclusions

In summary, NMIBC is a common cancer with a high rate of recurrence when treated with the standard treatment modalities. This results in patients requiring frequent invasive surveillance for many years after initial diagnosis. This can be burdensome to patients and costly for healthcare systems. When patients do not respond appropriately to standard therapies for NMIBC, cystectomy is the final, definitive treatment option. Not all patients are fit to undergo or willing to undergo cystectomy, and for those who do, it can be highly morbid. Additionally, there has been a nationwide shortage of one of the gold-standard treatments for NMIBC, BCG. For those reasons, the development of novel treatment modalities is essential. This review highlights novel, bladder-sparing treatment options and ongoing clinical trials for patients with NMIBC, categorized by risk-stratification and BCG-exposure status. In the past 10 years, there has been a significant expansion in both the volume of clinical trial offerings for patients with NMIBC and the FDA-approved therapies for BCG-refractory NMIBC. While we have presented a thorough review of published and ongoing studies, a limitation of this review is that the ongoing clinical trials discussed are limited to those occurring in the United States. Future review articles could emphasize the current trials underway in other countries. An underlying theme amongst research in this domain is limiting or excluding the use of BCG. There has been an ongoing shift towards increased usage of intravesical chemotherapy, specifically with gemcitabine/docetaxel, as BCG has become more difficult to obtain. In the future, we expect more research investigating increased options for intravesical chemotherapy and the continued development of novel technology for the delivery of intravesical therapy. Additionally, as there have been advances from a translational research standpoint and further information has been gathered regarding NMIBC at the molecular level, the options for targeted therapies will continue to increase. While radical cystectomy remains the gold standard for very high-risk or BCG-unresponsive disease, we see the promise that the options for patients unwilling or unfit for cystectomy will continue to increase.

## Figures and Tables

**Table 1 cancers-16-01843-t001:** Active Bacillus Calmette–Guérin naïve low-, intermediate-, and high-risk non-muscle invasive bladder cancer clinical trials ongoing within the United States.

Study Name	Study ID	Study Status	Study Size	Phase	NMIBC Risk	BCG Status	Therapeutic Agents
ADVANCED-2	NCT05951179	Recruiting	102	Phase Ib/II: Single Arm	High	Naïve, Unresponsive	TARA-002
BRIDGE	NCT05538663	Recruiting	870	Phase IIIb: Randomized Arm 1: Gemcitabine + Docetaxel Arm 2: BCG	High	Naïve	Gemcitabine, Docetaxel, BCG
CREST	NCT04165317	Active, not recruiting	1070 *	Phase III: Cohort 1: BCG-Naïve Arm A: PF-06801591 + BCG (induction + maintenance) Arm B: PF-06801591 + BCG (induction only) Arm C: BCG Cohort 2: BCG-Unresponsive Arm 1: BCG-Unresponsive CIS treated with PF-06801591 Arm 2: BCG-Unresponsive papillary-only disease treated with PF-06801591	High	Naïve, Unresponsive	Sasanlimab
EVER	NCT05037279	Not recruiting yet	540	Phase III: Randomized Arm 1: Verity-BCG Arm 2: OncoTICE	Intermediate, High	Naïve	BCG:Strain Russian BCG-I, BCG:Strain TICE
GEMDOCE	NCT04386746	Active, not recruiting	27 *	Phase II: Single Arm	Any	Naïve	Gemcitabine, Docetaxel
J18158	NCT03914794	Recruiting	43	Phase II: Single Arm	Low, Intermediate	Naïve	Pemigatinib
KEYNOTE-676	NCT03711032	Recruiting	1405	Phase III: Randomized Cohort 1: Arm 1: BCG plus Pembrolizumab: Post-induction Arm 2: BCG Monotherapy: Post-induction Cohort 2: Arm 1: BCG plus Pembrolizumab: BCG-Naïve -Reduced Maintenance Arm 2: BCG plus Pembrolizumab: BCG-Naïve -Full Maintenance Arm 3: BCG Monotherapy: BCG-Naïve	High	Naïve, Recurrent	Pembrolizumab, BCG
LEGEND	NCT04752722	Recruiting	222	Phase I/II Phase I: Single-Arm Phase II: Single-Arm	High	Naïve, Unresponsive	EG-70 (non-viral gene therapy)
PATAPSCO	NCT05943106	Recruiting	100	Phase IIIb: Single Arm	High	Naïve	Durvalumab, BCG
PIVOT-006	NCT06111235	Recruiting	426	Phase III: Randomized Arm A: Cretostimogene after TURBT Arm B: Observation after TURBT	Intermediate	?	Cretostimogene Grenadenorepvec, n-dodecyl-B-D-maltoside
SunRISe-3	NCT05714202	Recruiting	1050	Phase III: Randomized Arm A: TAR-200 + Cetrelimab Arm B: BCG Vesiculture Arm C: TAR-200 Alone	High	Naïve	TAR-200, cetrelimab, BCG Vesiculture

* Actual.

**Table 2 cancers-16-01843-t002:** Active Bacillus Calmette–Guérin unresponsive low-, intermediate-, and high-risk non-muscle invasive bladder cancer clinical trials ongoing within the United.

Study Name	Study ID	Study Status	Study Size	Phase	NMIBC Risk	BCG Status	Therapeutic Agents
ADVANCED-2	NCT05951179	Recruiting	102	Phase Ib/II: Single Arm	High	Naïve, Unresponsive	TARA-002
BOND-003	NCT04452591	Recruiting	110	Phase III: Single-Arm	High	Unresponsive	CG0070, n-dodecyl-B-D-maltoside
CORE-001	NCT04387461	Active, not recruiting	35 *	Phase II: Single-Arm	High	Unresponsive	CG0070, Pembrolizumab Injection, n-dodecyl-B-D-maltoside
CREST	NCT04165317	Active, not recruiting	1070 *	Phase III: Cohort 1: BCG-Naïve Arm A: PF-06801591 + BCG (induction+maintenance) Arm B: PF-06801591 + BCG (induction only) Arm C: BCG Cohort 2: BCG-Unresponsive Arm 1: BCG-Unresponsive CIS treated with PF-06801591 Arm 2: BCG-Unresponsive papillary-only disease treated with PF-06801591	High	Naïve, Unresponsive	Sasanlimab
ENVISION	NCT05243550	Active, not recruiting	220	Phase III: Single-Arm	High	≤24 months, but not nonresponsive	UGN-102
J18158	NCT03914794	Recruiting	43	Phase II: Single-Arm	Low, Intermediate	Unresponsive	Pemigatinib
KEYNOTE-057	NCT02625961	Recruiting	320	Phase II: Single-Arm Randomized	High	Unresponsive	Pembrolizumab, Pembrolizumab/vibostolimab, Favezelimab/pembrolizumab
SunRISe-1	NCT04640623	Recruiting	200	Phase II: Randomized Cohort 1: TAR-200 and Cetrelimab Cohort 2: TAR-200 Cohort 3: Cetrelimab Cohort 4: TAR-200 (Participants with Papillary Disease only)	High	Unresponsive	TAR-200, cetrelimab
SunRISe-5	NCT06211764	Not recruiting yet	250	Phase III: Randomized Arm 1: TAR-200 Arm 2: MMC or Gemcitabine	High	Unresponsive	TAR-200, Gemcitabine, Mitomycin C
THOR-2	NCT04172675	Active, not recruiting	107 *	Phase II: Randomized Cohort 1: high-risk NMIBC presenting as papillary tumor only (CIS, absent) Arm1: Erdafitinib Arm 2: Investigators Choice Cohort 2: high-risk, BCG- unresponsive treated w/ Erdafitinib Cohort 3: intermediate-risk NMIBC presenting as a papillary disease only treated w/Erdafitinib	High	Unresponsive	Erdafitinib, Gemcitabine, Mitomycin C
QUILT-3.032	NCT03022825	Recruiting	190	Phase II/III: Single Arm	High	Unresponsive	N-803 and BCG

* Actual.

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
