# Peer review of "Treatment Modalities for Non-Muscle Invasive Bladder Cancer: An Updated Review"

_cancers, 2024, doi:10.3390/cancers16101843_

Round 1

Reviewer 1 Report

Comments and Suggestions for Authors

The narrative review is very extensive and well written. It provides an interesting overview of the management of low risk non muscle invasive bladder cancer with treatment modalities. Overall, this is a well written paper.

In the introduction you mention the AUA guidelines (line 39). Please also provide info regarding the classification of EAU guidelines since are the most followed by worldwide urologists.

In methods explain how many researchers (oncologist? Urologists?) performed the literature search and their experience with bladder cancer treatment.

I would introduce a brief paragraph on Low risk bladder cancer follow-up / surveillance and the role of novel biomarkers in the treatment and follow-up decision (i.e. doi: 10.3390/ijerph19159648. - doi: 10.3390/diagnostics10010039. )

I would introduce the limitations of the study

Author Response

Thank you for your time taken to review this article.

Responses have all been incorporated in uploaded manuscript, summarized below:

In the introduction you mention the AUA guidelines (line 39). Please also provide info regarding the classification of EAU guidelines since are the most followed by worldwide urologists.

- Added a few sentences discussing EAU guidelines and how they compare to AUA guidelines.

In methods explain how many researchers (oncologist? Urologists?) performed the literature search and their experience with bladder cancer treatment.

- This has been added to methods

I would introduce a brief paragraph on Low risk bladder cancer follow-up / surveillance and the role of novel biomarkers in the treatment and follow-up decision (i.e. doi: 10.3390/ijerph19159648. - doi: 10.3390/diagnostics10010039. )

- Included information about development of urinary biomarkers in the section recommended. Also cited bladder cancer advocacy network reporting that patient survey responses specifically demonstrate that they value findings way to decrease discomfort associated with cystoscopic surveillance.

I would introduce the limitations of the study

- Limitations introduced in conclusions. Specifically that we only outlines ongoing US based clinical trials.

Reviewer 2 Report

Comments and Suggestions for Authors

This is a very comprehensive and timely review of the very important subject. Definitely, the treatment of NMIBC need to be analyzed and further developed. Review is well written and will be helpful for scientist that are studying NMIB and for practical urologists.

Author Response

Thank you for your response.

Reviewer 3 Report

Comments and Suggestions for Authors

This review article is well written and details treatment strategies for non-muscle invasive bladder cancer to date. 

Please check the following text.

P5, L220: We describe these emerging therapeutic strategies below

P10, L471: Study completion is expected in December2026 [41]

Author Response

Thank you for your comments. I have fixed the two sentences noted as they were missing punctuation. This will be reflected in the uploaded manuscript.